# Quality Control—A Stepchild in Quantitative Proteomics: A Case Study for the Human CSF Proteome

**DOI:** 10.3390/biom13030491

**Published:** 2023-03-07

**Authors:** Svitlana Rozanova, Julian Uszkoreit, Karin Schork, Bettina Serschnitzki, Martin Eisenacher, Lars Tönges, Katalin Barkovits-Boeddinghaus, Katrin Marcus

**Affiliations:** 1Medical Proteome Analysis, Center for Protein Diagnostics (ProDi), Ruhr University Bochum, 44801 Bochum, Germany; 2Medizinisches Proteom-Center, Medical Faculty, Ruhr University Bochum, 44801 Bochum, Germany; 3Department of Neurology, Ruhr-University Bochum at St Josef-Hospital, 44801 Bochum, Germany

**Keywords:** mass spectrometry, proteomics, label-free quantification, cerebrospinal fluid, FASP, in-solution digestion, quality control

## Abstract

Proteomic studies using mass spectrometry (MS)-based quantification are a main approach to the discovery of new biomarkers. However, a number of analytical conditions in front and during MS data acquisition can affect the accuracy of the obtained outcome. Therefore, comprehensive quality assessment of the acquired data plays a central role in quantitative proteomics, though, due to the immense complexity of MS data, it is often neglected. Here, we address practically the quality assessment of quantitative MS data, describing key steps for the evaluation, including the levels of raw data, identification and quantification. With this, four independent datasets from cerebrospinal fluid, an important biofluid for neurodegenerative disease biomarker studies, were assessed, demonstrating that sample processing-based differences are already reflected at all three levels but with varying impacts on the quality of the quantitative data. Specifically, we provide guidance to critically interpret the quality of MS data for quantitative proteomics. Moreover, we provide the free and open source quality control tool *MaCProQC*, enabling systematic, rapid and uncomplicated data comparison of raw data, identification and feature detection levels through defined quality metrics and a step-by-step quality control workflow.

## 1. Introduction

The label-free mass spectrometry (MS)-based protein quantification technique enables the detection of differences in protein patterns on a global scale. Today, it is used in a variety of applications in fundamental and medical research [1,2,3]. To uncover differentially abundant proteins as potential biomarkers in high-throughput screening, label-free bottom-up (peptide-based) quantitative proteomics can be used [4,5]. Although this approach is standard and is ubiquitously used in various scientific fields, several challenges still exist [6,7,8,9]. These include a variety of factors that have a major impact on the quality of quantitative MS data, which is often neglected in data analysis and interpretation.

For clinical samples, for instance, there is the factor of biological variance and sample quality. For cerebrospinal fluid (CSF), a body fluid that has an important role in liquid biopsies for the detection of biomarkers for neurodegenerative diseases due to its direct contact with the central nervous system [10,11,12,13,14], we have demonstrated that a large variability in individual proteins is present in healthy controls [10,11,12,13,14]. Hence, it is essential to address this variability in order to avoid a negative influence on protein quantification. Higher levels of alpha-synuclein in CSF, for example, a protein described as a possible molecular marker for Parkinson’s Disease (PD) [15], may result from contamination with blood [16] when CSF is collected. (Alpha-synuclein is one of the most abundant proteins in erythrocytes.) The level, as well as the possible origin, of the proteins must therefore be carefully evaluated within a quantitative proteome analysis of CSF.

On the analytical site, a label-free bottom-up MS quantitative approach comprises multiple steps [5], and each of them can introduce variances and lower the confidence of the obtained results [9]. The first step includes extraction of proteins, their denaturation, reduction, and alkylation and further digestion to peptides with proteases (typically trypsin). Here, variations can be introduced due to the ineffectiveness and/or irreproducibility of digestion or due to the introduction of unexpected variable modifications [17,18], which can later result in an increased number of unassigned peptide spectra [19,20,21]. Following sample preparation, peptides are usually separated with liquid chromatography (LC) based on their hydrophobicity. At this step, sample-dependent characteristics can affect accurate quantification, e.g., the presence of peptides with poor chromatographic behavior, overloading, cross-contamination due to sample carry-over, or the presence of contaminants [8,9]. In addition, the performance of the LC system as well as that of the MS system can lead to variations between samples and, thus, have an influence on the quality of the recorded data [22]. Monitoring of the chromatographic and mass spectrometer performance by using internal and/or external standards is therefore important [8,9]. Finally, the accuracy of the obtained findings is highly dependent on the bioinformatics handling of the MS data; i.e., the use of databases, spectral libraries, search engines together with applied search parameters and statistics as well as normalization [23,24,25,26].

Due to the various steps within an MS-based proteomics workflow, quality assessment of MS data is essential for comparative quantitative proteomic studies. Several important steps have already been taken by the proteomics community to standardize data quality control and data reports [27,28]. There are a number of tools available to assess the quality of MS raw data (e.g., MSQC [29], QuaMeter [30], Metriculator [31], SIMPATIQCO [32], PACOM [33], QCloud2 [34,35], SProCoP [36], QC-ART [37]), proteomic quantitative data (pmartR [38]), or data generated by the MaxQuant software [39]. Nevertheless, quality assessment of MS data is insufficiently considered or rarely addressed in publications. This could be due to either the lack of easy-to-use tools or varying perceptions of which matrices to consider when assessing the quality of MS data. The quality control tools provide varying degrees of quality assessment at the level of raw data (e.g., number of precursors (MS1) and fragment (MS2) ion scans, intensity of chromatographic peaks, charge states, etc.) and few are combined with peptide identifications to assess identification level parameters (e.g., number of missed cleavages, number of identifications). MSQC [29], Metriculator [31], and SIMPATIQCO [32] assess chromatographic performance, spray stability, total ion current, and identification efficiency; however, they are limited to one vendor data format. QuaMeter allows analysis similar to MSQC, and additionally allows different vendor formats [30], but due to the high number of metrics analysis outcomes, it is too sophisticated for the unexperienced user. SProCoP and QCloud2 do not require identification and are designed to evaluate MS-LC system performance using internal and/or external standards [34,35,36]; however, they do not allow quality control of the study samples. QC-ART enables real-time scoring of LC-MS performance along with thresholds to identify inconsistent runs that may require further evaluation [37], but it does not give any specific information regarding identification quality. PACOM is able to compare numerous proteomics experiments to check the completeness and quality of proteomic data, but only at the identification level [33]. PmartR is mainly designed to assess the quality of quantification and normalization [38], and does not consider LC-MS performance and raw data quality. Still, it is not clear whether quality control at the raw data and/or identification level is sufficient to ensure the quality of quantification as none of the tools include all the levels of assessment.

This work therefore explicitly addresses the comprehensive assessment of quantification quality in bottom-up proteomic studies using the example of CSF. We describe here the most important data quality metrics and introduce an open source quality control (QC) tool, *MaCProQC*, enabling rapid and easy data comparison at all levels (raw data, identification, and quantitative feature) in the same run through pre-defined quality metrics.

## 2. Materials and Methods

### 2.1. Samples

The CSF sample, collected by routine lumbar puncture from a patient with a neurodegenerative disease, was randomly selected from the Bochum cohort (Bochum cohort; institutional review board (IRB) number: # 17-6119), processed as previously described [40], and stored at −80 °C until subsequent analysis. Before further preparation steps, samples were thawed at room temperature and the protein concentration was measured according to Bradford [41] using Bio-Rad Protein Assay Dye Reagent (Bio-Rad Laboratories, USA, Cat# 5000006).

Four sample preparation approaches were used: standard and rapid in-solution digestions as well as standard and rapid filter-aided sample preparations (FASP). Each sample preparation protocol was run in five technical replicates (Appendix A).

### 2.2. Standard In-Solution Digestion

The standard in-solution digestion was carried out as described earlier, with minimal changes [40]. Briefly, 25 µg of CSF protein was mixed 1:1 *v*/*v* with 0.2% RapiGest (Waters, Eschborn, Germany). Proteins were reduced with 5 mM dithiotreitol (DDT) and alkylated with 15 mM iodoacetamide (IAA). Protein digestion was performed with trypsin (Serva, Heidelberg, Germany) at an enzyme-to-substrate ratio of 1:50 at 37 °C overnight. The digestion was stopped and RapiGest precipitated by acidification with trifluoroacetic acid (TFA).

### 2.3. Rapid In-Solution Digestion

For the rapid in-solution digestion of CSF samples, a Rapid Digestion Trypsin kit (Promega, Madison, WI, USA, Cat#VA1060) was used according to the manufacturer’s protocol. Briefly, 25 μg of CSF protein was mixed with 3 sample volumes of the Rapid Digest Buffer. Disulfide bond reduction was carried out via 45 min incubation at 37 °C with 2 mM final concentration of Tris(2-carboxyethyl)phosphine (TCEP). In order to alkylate free thiols, the protein solution was incubated for 60 min at room temperature in darkness with 5 mM IAA. Proteins were digested via 60 min incubation with the rapid trypsin (enzyme-to-substrate ratio 1:10) at 70 °C on a shaker at 450 rpm. The tryptic digestion was stopped by adding TFA to a final concentration of 0.5%.

### 2.4. Standard Filter-Aided Sample Preparation (FASP)

For the filter-aided sample preparation (FASP) [42], 200 µL of 8 M Urea was added to the 25 µg of the sample protein. After 60 min incubation at 37 °C with 10 mM final concentration DTT, proteins were alkylated for 30 min at room temperature in darkness with 5 mM iodoacetamide (IAA). Before digestion, samples were applied to the Vivacon filter (0.5 mL, 10.000 MWCO, Sartorius Stedium Lab Ltd., Stoneouse, UK) and centrifuged for 15 min at 12,000× *g*. Further samples were washed twice with 200 µL of 8 M Urea and 3 times with 200 µL of ammonium bicarbonate (Ambic), with 20 min centrifugation at 12,000× *g* in between. After washing, each filter was transferred to a new tube and the sample was resuspended with 20 µL of 50 mM Ambic. Proteins were digested via incubation with trypsin (Serva, Heidelberg, Germany) (enzyme-to-substrate ratio 1:50) at 37 °C overnight. Peptide elution was carried out by adding 50 µL of 50 mM Ambic and 15 min centrifugation at 14,000× *g* at 18 °C. Digestion was stopped with 0.5% TFA.

### 2.5. Rapid FASP

For the Rapid FASP, samples were reduced and alkylated as described above (Section 2, Section 2.3). Before digestion, samples were applied to the Vivacon filters and centrifuged for 15 min at 12,000× *g*. Further samples were washed 3 times for 5 min with 200 µL of Rapid Digest Buffer, with 15 min centrifugation at 12,000× *g* in between. After washing, each filter was transferred to a new tube and the sample was resuspended in 20 µL of Rapid Digest Buffer. Proteins were digested via 1 h incubation with rapid Promega trypsin (enzyme to substrate ratio 1:10) at 70 °C. Peptide elution was carried out by adding 50 µL of 50 mM Ambic and 15 min centrifugation at 14,000× *g* at 18 °C. Digestion was stopped by 0.5% TFA.

### 2.6. Label-Free NanoLC-MS with DDA Aquisition

The peptide concentration in the digested samples was determined by amino acid analysis (AAA) as described by Plum [43] and May et al. [44]. According to AAA, 500 ng per sample was taken for nanoLC-MS analysis and analyzed as previously described [43,44]. Briefly, extracted peptides were first injected and pre-concentrated with an UltiMate™ 3000 RSLCnano system (Thermo Fisher Scientific Inc., Waltham, MA, USA) using a trap column (Acclaim PepMap 100, 300 μm × 5 mm, C18, 5 μm, 100 Å; flow rate 30 μL/min). Subsequently, the peptides were separated on the analytical column (Acclaim PepMap RSLC, 75 μm × 50 cm, nano Viper, C18, 2 μm, 100 Å) by a gradient from 5–40% solvent B over 98 min (solvent A: 0.1% FA in water; solvent B: 0.1% FA, 84% Acetonitrile in water; flow rate 400 nL/min; column oven temperature 60 °C). Mass spectrometry analysis was performed as earlier described, with some modifications [40,45,46,47]. Separated peptides were ionized by electrospray ionization (ESI) and injected into an Orbitrap Fusion™ Lumos™ Tribrid™ Mass Spectrometer (Thermo Fisher Scientific, Bremen, Germany). The capillary temperature was set to 275 °C and the spray voltage to 1500 V. The lock mass polydimethylcyclosiloxane (445.120 m/z) was used for internal recalibration. The instrument was operated in data-dependent acquisition (DDA) mode with 2 s cycle time, performing HCD fragmentation at 28% NCE. The mass range for MS1 measurements was set to 350–1400 m/z with an orbitrap resolution of 120,000 at 200 m/z (AGC 3e6, 80 ms maximum injection time, 1.3 m/z wide isolation window, 30 s dynamic exclusion). The fragment analysis was performed in an orbitrap mass analyzer with a resolution of 30,000 at 200 m/z (AGC 3e6, 80 ms maximum injection time). The mass spectrometry proteomics data from the DDA analysis have been deposited with the ProteomeXchange Consortium via the PRIDE partner repository [48] with the dataset identifier PXD037650 and 10.6019/PXD037650.

For the evaluation of nanoLC-MS system performance, a complex external standard (digest of the human cell line A549) was measured at the beginning, in the middle, and at the end of the study series. For these control runs, column pressure profiles, spray stability, symmetry, width, and intensity of chromatographic peaks, LC pressure profiles, and retention time stability were checked. Moreover, the mass spectrometer performance was evaluated by the number of acquired precursor and fragment ion spectra as well as by the number of identified peptides and protein groups.

### 2.7. Data Analysis

Raw files were analyzed using an in-house developed quality control tool, *MaCProQC* (Mass speCtrometry Proteomics QC) (see in Appendix A and https://doi.org/10.48546/WORKFLOWHUB.WORKFLOW.343.1) (accessed on 7 February 2023), implemented into a KNIME [49] workflow and making use of the Mascot search engine [50] (www.matrixscience.com, accessed on 7 February 2023), PIA [51,52], and OpenMS [53]. Raw files were processed by the workflow, and the chromatographic profiles of total ion chromatograms (TICs), precursor, and fragment ions, and the number of precursor and fragment ions as well as their charge states, were analyzed. Further, evaluation at the identification level was performed. The spectra were identified using Mascot with the human reference proteome (UP000005640) of the UniProtKB [54] database (release 2021_11) using the following parameters: enzyme = trypsin, maximum missed cleavages = 3, variable modifications = methionine oxidation, fixed modification = cysteine carbamidomethylation, precursor mass tolerance = 5 ppm, and fragment mass tolerance = 20 mmu. PSM identification was performed using a decoy-based false discovery rate of 0.01. Peptides were inferred on the amino acid sequence using PIA, and protein groups were inferred and filtered (0.01 FDR) using PIA’s “Spectrum Extractor” approach, which utilizes only one peptide identification per spectrum. Principal component analysis based on the raw data features and all the data features (raw data and identifications) were performed and visualized. Additionally, potentially overlooked modifications were analyzed using Mascot’s error-tolerant search.

In order to check for additional overlooked peptide modifications, a Mascot error-tolerant search (www.matrixscience.com, accessed on 7 February 2023) was conducted against the human reference proteome (UP000005640) of the UniProtKB database (release 2021_11) using the following parameters: enzyme = trypsin, missed cleavage = 2, precursor mass tolerance = 5 ppm, fragment mass tolerance = 20 mmu.

As a further evaluation, peptides and proteins were quantified using MaxQuant (v.2.0.3.1, https://maxquant.org/, accessed on 7 February 2023). Spectra were searched against the human reference proteome (UP000005640) of the UniProtKB database (release 2021_11) using the following parameters: enzyme = trypsin, maximum missed cleavages = 2. Methionine oxidation was set as variable modifications; cysteine carbamidomethylation as fixed. Instrument settings were left as default. PSM identification was performed using a reversed decoy-based false discovery rate of 0.01. The minimum number of peptides and razor peptides for protein identification was 1; the minimum number of unique peptides was 0. Protein identification was performed at a protein false discovery rate of 0.01. The “match between runs” option was disabled. Proteins were quantified with intensities and using MaxQuant label-free quantification (LFQ) [55], with at least one unique peptide at the minimum ratio count of 2. The “classic” normalization [55] was applied to the LFQs. MaxQuant data analysis was performed using Perseus [56] and R version 4.1.0 (R Core Team, 2021).

Moreover, data normalization quality analysis was conducted and graphics were created using R version 4.1.0 (R Core Team, 2021), using the ggplot2 package version 3.3.5 [57,58]. The analysis was performed separately on protein groups and peptide levels. The proteinGroups.txt and peptides.txt files from the MaxQuant output were loaded into R and raw intensities and LFQs were used for further analysis. Intensity and LFQ values of 0 were considered as missing values and substituted by NA. Protein group and peptide level data (raw intensities) were log2-transformed and then normalized using R package limma (Version 3.47.16) [59] with LOESS normalization [60]. The quality of the normalization was assessed using boxplots and PCA plots. For the PCA plots, only proteins without missing values were used. In addition, MA-Plots, scatter plots of log2 fold changes (M) versus the average intensity (A), were generated for the replicates of the datasets [25] and the local regression was assessed. The R-scripts for the quantification quality analysis are available at https://github.com/mpc-bioinformatics/QC_Quant (v1_2) (accessed on 7 February 2023).

## 3. Results

The evaluation of quantitative proteomics data is important to obtain consistent results. However, quality control (QC) is mostly not performed at the quantitative level or is even omitted. Therefore, we aimed to evaluate and establish a systematic and comprehensive quality control (QC) workflow for quantitative proteomics data. This universal workflow was comparatively applied to different quantitative proteomic datasets obtained for CSF, an important specimen for neurological diseases that reflects disease-related protein changes due to its direct contact with the brain. In order to generate independent and different CSF proteome datasets, four different digestion strategies were used, while the MS-based measurements were performed equally. The subsequent data analysis and evaluation was divided into three main stages: (1) raw data, (2) identification, and (3) quantification quality assessment. Each of these stages includes a set of metrics to capture variations caused by sample handling/preparation, nanoLC-MS performance, and/or data processing. For comprehensive evaluation of the MS data for stages 1 and 2, a quality control tool, *MaCProQC*, was developed, enabling fast and straightforward data processing and comparison of raw data and identification feature detection level. The tool is provided via WorkflowHub (https://doi.org/10.5281/zenodo.4605654 accessed on 7 February 2023) at [https://doi.org/10.48546/WORKFLOWHUB.WORKFLOW.343.1] and further described in the Appendix A. Quantitative data were generated with MaxQuant, and the quality of the label-free quantification was assessed within stage 3 using a custom workflow based on R-scripts, available at https://github.com/mpc-bioinformatics/QC_Quant (accessed on 7 February 2023).

### 3.1. Impact of Sample Preparation on Peptide Recovery

CSF datasets we generated used four digestion strategies, including standard in-solution digestion, rapid in-solution digestion, standard filter-aided sample preparation (FASP), and rapid FASP, assuming different sample preparations result in different peptide recovery after digestion, leading to variations at the peptide and protein levels. To ensure that the same amount of peptides was analyzed using MS for all the samples, amino acid analysis (AAA) [44] had been performed for peptide quantification. The peptide quantification results already confirmed that the different digestion methods vary in sample recovery. Briefly, both FASP methods, in contrast to in-solution digestion approaches, resulted in sample losses during digestion, with recoveries of 46.6% and 22.0% for standard FASP and rapid FASP, respectively (Appendix A). Moreover, for the standard FASP, a relatively high variance (19.7%) between replicates was observed. In comparison, for in-solution digestions, this parameter did not exceed 4.6% (Appendix A).

### 3.2. Raw Data Quality Assessment (Stage 1)

In the first assessment stage, which involved the evaluation of the raw data quality, the following metrics were considered:Number of acquired MS1 and MS2 spectra;Ion chromatogram profile;Distribution of precursor charge states.

#### 3.2.1. NanoLC-MS: Total Number of Scan Events

The total numbers of MS1 and MS2 scans are important parameters for data quality evaluation. Comparing the number of MS1 and MS2 spectra within the different datasets, the coefficient of variation (CV) ranged from 0.7% to 6.7% in MS1 and from 0.8% to 5.9% in MS2. Here, in particular, the highest CV for both levels was observed for rapid FASP (Figure 1a,b), indicating less reproducible processing of the samples. The lowest variation for both MS1 and MS2 was observed for the rapid in-solution digestion, with 0.7% and 0.8%, respectively.

#### 3.2.2. NanoLC-MS: Ion Chromatogram and Charge State Distribution Evaluation

In order to enable evaluation of sample processing and stability of the LC-MS system, the chromatographic profiles of the TICs, precursor, and fragment ions were examined. The first metric analyzed was the accumulated TICs, which provide a rough assessment of the quantitative similarity of the data at the MS1 level and confirm that the same amount of sample was introduced into the column. The CVs for all four different datasets were between 10–20% (Figure 2). The accumulated TICs showed different numerical values, with the lowest one for the standard in-solution digestion and the highest one for the rapid FASP. The observed deviations for the four datasets indicate differences in the precursor composition of the samples.

As a next step, the peak intensity distribution along the retention time (RT) was examined based on the TIC quantile profile. The assessed quantiles reflect fractions of the total RT referring to the first, second, third, and fourth quarter of the total run’s intensity (i.e., the quartiles). This metric enables a numerical comparison of chromatogram intensity distribution without any visual inspection. In our datasets, the metric values were comparable, besides one outlier for the standard in-solution digestion dataset and a slight chromatogram intensity variation (seen as a shift of the third and fourth quantiles) for the rapid FASP (Figure 3a; for the visual inspection, see Appendix A). Comparison of MS1 and MS2 ion intensities revealed no severe difference (Appendix A).

Significant differences among the datasets were observed for the distribution of precursor ion charge states. Overall, the rapid in-solution digestion has the highest deviation in the distribution compared to the others, with an enhanced number of highly charged ions. Both in-solution digestion approaches showed a more stable precursor charge distribution for the replicates (Figure 3b).

To visualize the results of all raw level metrics (Appendix A: MaCProQC—Mass spectrometry proteomics quality control workflow) in a two-dimensional plot and to assess the inter- and intragroup variability, principal component analysis (PCA) was performed. Overall, the PCA plot revealed that the generated datasets differ from each other, while the highest intragroup variability occurred in the rapid FASP dataset (Figure 4). A closer look at the data indicated that variations at the MS2 level (number of spectra) have the greatest impact on the first principal component (PC1) (Appendix A).

### 3.3. Identification Quality Assessment

The second main step of the QC workflow comprised qualitative evaluation of the identification level, which includes the following metrics:Number of FDR-controlled peptide spectrum matches (PSMs), peptides, and protein groups (PGs);Percentage of missed cleavage states.

#### 3.3.1. Number of Identified Peptides/Protein Groups (PGs)

Among the datasets, the highest mean values for the peptide and protein group (PG) identifications were observed for the standard FASP (Figure 5). The rapid FASP resulted in the lowest number of identified peptides and also a high variation within this metric, which is in concordance with the above-discussed data for the number of acquired MS2 spectra (see Section 3.2.1).

#### 3.3.2. Peptide Identification Quality Assessment: Missed Cleavages

Next, we investigated the effectiveness and the reproducibility of the digestion. Within the CSF datasets, the percentage of PSMs for peptides carrying missed cleavage sites ranged from 10% to 22%. The lowest value was observed for the rapid FASP and the highest for the rapid in-solution digestion. A higher percentage of missed cleavages explains the increased number of highly charged ions for the rapid in-solution digestion (Figure 3) discussed above. The number of missed cleavages and the distribution of fractions showing 0–3 missed cleavages varied significantly for the standard FASP, while it was stable for the other datasets (Figure 6). These inconsistencies are also in line with the results of the charge state distribution for the standard FASP. 

To visualize the outcome from the first two stages of data quality assessment, the PCA based on raw data and identification metrics was calculated (Figure 7). Overall, the PCA profile was comparable to the one based on raw data features in Figure 4, with a slightly higher intergroup separation for the standard FASP. Detailed data examination revealed that standard FASP clustering is mainly affected by differences in PSM distribution depending on their charge states (Appendix A). In summary, including the identification metrics, it was reconfirmed that among the four datasets, the rapid FASP dataset was the most divergent, which is most closely associated with the observed variances at the MS1 and MS2 levels, which provide a basis for the identification. 

In our study, samples were measured in a non-randomized manner. To exclude the possibility that the observed differences were due to a batch effect, four technical replicates were randomly re-measured for each CSF dataset. The data obtained were analyzed using MaCProQC and showed that the observed variances were not due to a batch effect. It was confirmed that the rapid FASP was significantly different from the other three datasets (data not shown).

### 3.4. QC for Label-Free Quantification

To verify that the variations detected by the MaCProQC tool were also reflected at the quantitative level, the quality of label-free quantification was assessed for all generated datasets using the following metrics:Number of quantified peptides/protein groups and data completeness;Similarity of the replicates based on clustering;Efficiency of data normalization.

#### 3.4.1. Number of Quantified Peptides/Protein Groups

To generate and access quantitative data, all CSF datasets were analyzed together using MaxQuant. Based on the intensity values, the highest total number of quantified peptides as well as protein groups among the datasets was observed for standard FASP (4678 and 615, respectively) (Figure 8a,b) and the lowest for rapid FASP (3080 and 496, respectively). However, on the replicate level, both FASP datasets showed high variation in the number of quantified peptides, while the rapid FASP exhibited the highest variation in the number of quantified PGs (Figure 8a,b). This result is also in line with the variability observed in the numbers of MS1 and MS2 scans for the rapid FASP dataset. Such variances usually impact the data completeness (i.e., fraction of quantifications per dataset compared to quantifications in all replicates of a dataset), which could be confirmed within our datasets. Although the standard FASP dataset had the highest total number of quantified peptides and PGs, these numbers were reduced by 51% and 52%, respectively, when only overlapping identifications were considered (Appendix A, Figure 8a). Thus, the standard FASP dataset showed comparable completeness with respect to the rapid in-solution digestion dataset, with 2299 and 2275 peptides, or 451 and 428 PGs, respectively. These two datasets showed the best data completeness among the CSF datasets, while the rapid FASP had the lowest one: 1493 for peptides and 277 for the PGs (Appendix A).

In order to evaluate the observed variance and low data completeness for the rapid FASP approach, all the datasets were assessed regarding physicochemical characteristics and the nature of peptides and their corresponding proteins, including hydrophobicity, sequence length (number of amino acids), molecular weight, and abundancy (data are not shown). None of the checked characteristics could explain the discrepancies, except that more missing values were observed for the low abundant proteins. Peptide and protein ID comparison for different datasets showed high overlapping (data are not shown).

In addition, we analyzed the datasets using a Mascot error-tolerant search to check if there were any previously unaccounted modifications in the datasets. As expected, carbamidomethylation of Cys and oxidation of Met were detected in all of the samples, showing comparable total matches for the replicates from the same dataset. For the standard FASP additionally, highly varying carbamylation of N-terminus was detected (CV = 63.6%) (Appendix A).

#### 3.4.2. Replicate Similarity Based on Correlation

In order to evaluate the quantitative data in regard to similarities among and within the CSF datasets, Pearson correlation coefficient analysis for peptide (Figure 9a) and PG (Figure 9b) intensities was performed. The analysis revealed that the generated CSF datasets differed from each other at both peptide and PG levels, while the difference at the peptide level was more significant than at the PG level. Overall, rapid FASP resulted in the most divergent dataset, indicated by low Pearson correlation coefficients between the rapid FASP and other dataset replicates (0.74–0.86 for peptides and 0.82–0.94 for PGs). In addition, among the four datasets, the rapid FASP dataset showed the lowest Pearson correlation coefficient between the individual replicates, ranging from 0.88 to 0.97 for peptides and from 0.82 to 0.97 for PGs. The standard and rapid in-solution digestion as well as standard FASP datasets showed similar correlation for the respective replicates at the PG level; however, at the peptide level, the standard FASP showed higher variation (from 0.90 to 0.96) compared to the in-solution digestion replicates.

#### 3.4.3. Assessment of Data Normalization

Data normalization is a standard step used within a quantitative proteomics workflow in order to reduce the technical variability of the data. Therefore, at the next stage, the impact of two different normalization strategies, MaxQuant LFQs and LOESS normalization (applied to the log2-transformed raw intensities), was evaluated for each of the CSF datasets (in Appendix A: “Normalization_loess_proteins.xlsx”). Compared to the LFQ-based normalization implemented in the MaxQuant workflow, LOESS normalization has the advantage that the datasets are normalized separately after the joint search. The MaxQuant normalization of LFQs is an inseparable part of the commonly used label-free quantification workflow implemented in MaxQuant and cannot be applied separately to each dataset within one search run. Since the studied datasets represent different peptide compositions, their joint normalization is questionable. To apply the MaxQuant normalization separately, each dataset has to be searched separately. This can lead to different protein inference and, as a result, to varying protein group lists, which makes it hard to compare the data.

The boxplots of CSF datasets without normalization showed comparable PG intensity distribution, with variations between the replicates for all of the datasets (Appendix A). These discrepancies were reduced by both MaxQuant LFQ and LOESS normalization approaches in all the datasets except the rapid FASP (Appendix A). Similar results were obtained with the PCA (Figure 10) and MA plots (Appendix A). Overall, rapid FASP was the most divergent dataset, as indicated by a larger distance between its replicates at the PC1 level compared to other datasets. Both LOESS and MaxQuant LFQ normalization worked comparably well, improving the replicate clustering for all the datasets except for the rapid FASP (Figure 10b,c).

In summary, stepwise quality assurance allowed us to determine the rapid FASP as the least reproducible protocol and thus unsuitable for the quantitative proteomic studies. Both in-solution digestion protocols showed the best quality of quantification, although standard in-solution digestion yielded less quantified PGs compared to the rapid in-solution digestion (Appendix A).

## 4. Discussion

The quality of quantitative proteomics data is critical for the reliability and validity of the research findings and strongly depends on pre-analytical sample processing as well as the reproducibility and robustness of the analytical LC-MS platform. Several approaches have been reported to assess the quality of proteomics MS data [30,31,32,39] which mainly consider aspects like the total number of scan events, the distribution of ion charge states, the overfill or underfill ratio of the automatic gain control, the number of identified PSMs, peptides and PGs, as well as the percentage of peptides with missed cleavages [30,31,32,39]. These parameters are very important for data quality assessment but do not comprehensively reflect the quality of the quantitative data. For example, the number of identified peptides and/or PGs in comparative samples may be similar, but the overlap of identical identified peptides and/or PGs may be very small, leading to low data completeness and, consequently, a high number of missing values in the quantitative data.

In this study, a systematic and comprehensive workflow for quality evaluation of label-free quantitative MS data was demonstrated utilizing the free and open access *MaCProQC* tool, which is easy to use even for inexperienced users and allows the attaining of a comprehensive qualitative data quality assessment. The *MaCProQC* tool implements various metrics to evaluate MS data, allowing for assessment at the raw data and identification levels. The advantage of the tool is that it also provides a general overview of data quality across raw data and identification metrics by performing principal component analysis, revealing inconsistent runs or systematic quality shifts. Individual metrics can be further reviewed to determine the cause of inconsistency and/or low data quality in order to avoid them in the future. Evaluation of the raw data gives a first impression of the data quality: the number of MS1 and MS2 spectra allows a rough evaluation of the peptide identification and quantification rate [29]; evaluation of chromatographic profiles is useful both for revealing inconsistencies in sample processing/preparation and for detecting instabilities in the LC system [9,29]; and the distribution of the precursor charge states allows the prediction of digestion efficiency and reproducibility. Quality evaluation at the identification level included some more profound information on data quality: numbers of PSMs, peptides, PGs, and the variations in these values, as well as the efficiency and reproducibility of protein digestion, earlier shown to be highly important for protein accurate quantification [28,61]. These metrics enabled the detection of differences within and between the four CSF datasets, with the rapid FASP approach showing the highest overall differences in comparison. In addition, in order to link the results obtained with the *MaCProQC* to quantitative data, the quality of label-free quantification was assessed for each dataset by the number of quantified peptides/PGs and level of data completeness, the similarity of the replicates based on clustering, and, finally, the efficacy of normalization.

Quantification quality analysis showed that the rapid FASP had the lowest data completeness compared to the other datasets. The *MaCProQC* tool reflected this outcome at both the raw data and identification levels by detecting the highest number of MS1 and lowest number of MS2 spectra acquired for the rapid FASP, indicating lower sampling, and by identifying the lowest number of peptides and PGs compared to other datasets. The highest variance in the number of quantified peptides and PGs and the lowest Pearson correlation coefficients between replicates were also observed in the rapid FASP. These inconsistencies were reflected in several *MaCProQC* metrics: high variation in MS1 and MS2 scan numbers, shifts in the ion chromatogram and charge state distributions, and high variability in the number of peptides and PGs identified. Neither physicochemical properties nor the nature of the peptides and corresponding proteins could explain the discrepancies for the rapid FASP. Previous qualitative differences for different cell line preparations were reported by Varnavides et al. [62]. In our study, peptide and protein ID comparison for different datasets showed high overlap, thus no qualitative differences in protein recovery from CSF were observed. The reason for these different results possibly lies in the nature of the samples. A combination of two main aspects could be the reason for the observed divergence in the rapid FASP approach. On the one hand, time of digestion could be a limiting factor. Insufficient digestion, also demonstrated for rapid in-solution digestion, may limit the passage of longer peptides through the filter device. On the other hand, it may be due to the irreproducible recovery of peptides from the filter, which has also been observed for standard FASP. Most likely, the irreproducible recovery for FASPs is attributed to the earlier reported inhomogeneous spreading of proteins through the filter membrane during washing and digestion [58].

The standard FASP, together with the rapid in-solution digestion protocol, showed the highest data completeness, however with more missing values and lower Pearson correlation coefficients for the replicates on the peptide level. The MaCProQC tool reflected the quantitative deviations for the standard FASP through the detection of high variations in the charge state distributions of the precursors and shifts in the distribution of PSMs based on number of missed cleavage sites. Both metrics are associated with non-reproducible digestion with standard FASP, due to either use of urea in sample preparation [63,64,65] and/or irreproducible recovery of the peptides from the filter. Thus, a highly variable (CV = 63.6%) carbamylation of the N terminus between replicates was detected by a Mascot error-tolerant search (Appendix A). This result could be due to non-reproducible and insufficient washing of proteins from urea, as the fixed-angle rotor centrifuges used in FASP apply pressure to a specific membrane site, resulting in inconsistent washing of protein concentration at one site [58]. Irreproducible carbamylation and digestion could be a reason for the lower-standard quality demonstrated by the MaCProQC tool (Figure 7) and quantitative cluster analysis (Figure 9). If the observed inconsistencies in peptide recovery are related to the filter itself, the reproducibility of the FASP protocol may be improved by the application of an automated positive-pressure approach [66].

According to our findings, both in-solution digestion protocols showed the best quantification quality as defined by Pearson correlation analysis. However, the *MaCProQC* tool detected a lower number of identified peptides and PGs for standard in-solution digestion and the highest percentage of missed cleavages (22%) for the rapid in-solution digestion protocol. Thus, for the discovery studies, the rapid in-solution protocol which enables the highest number of identifications at the highest quantification quality is recommended. Although the digestion procedure was reproducible and did not affect the accuracy of relative peptide and protein quantification, this factor could be a limitation for some studies, for example, when heavy peptides are spiked into the sample for absolute quantification. Therefore, we suggest the method of CSF digestion should be chosen based on the experimental design and the research question. This is also in agreement with the data reported by Varnavides et al., demonstrating that even for the same sample type, there is no single protocol that addresses all research questions due to qualitative and quantitative differences [62]. We also claim that for other sample types, the presented digestion strategies should still be tested and thoroughly evaluated, e.g., using *MaCProQC*.

Quantitative data normalization is commonly used to account for the bias introduced by experimental variation and to make the samples more comparable [25]. For reliable comparison of proteomic data, the bioinformatic processing (search and quantification) must be carried out in the same way (for all the samples in one analysis run). However, in the case of different dataset comparison—e.g., different preparations, differentiated versus non-differentiated cells, or whole tissue versus specific tissue regions—their separate normalization is required. Separate normalization of datasets in MaxQuant requires their processing to be performed separately, resulting in unequal identification and quantification. This challenge can be overcome with our approach, which involves post-processing of data using LOESS normalization, which is optimal for proteomic data [67]. In order to normalize the four datasets separately and further evaluate the effectiveness of data normalization, we developed an R-script, which we present here. The data can be LOESS-normalized and then subjected to normalization evaluation using the developed R-script. In our hands, both normalizations, the separate LOESS normalization of the raw intensities and the joint MaxQuant LFQ normalization, showed similar results. Nevertheless, we recommend using a separate normalization for those kinds of studies.

In our opinion, a QC combination via the assessment of both experimental samples and additional external standards (measured between study runs) provides a comprehensive control for conducting quantitative label-free proteome studies. QC of external standards allows for the distinguishing of problems related to the LC-MS system, while QC of acquired experimental data provides information on sample quality. Establishing thresholds for data quality for clinical studies is a difficult task because it depends on the type of sample, sample preparation method, HPLC system, columns, and MS instrument. However, it can be achieved for external standards and is already available with the tool in our laboratory. Thus, after a period of data collection, metric thresholds can be established in each individual laboratory for each LC-MS system, regardless of the external standard used there. The experience with QC metrics for CSF samples documented here will shape later discussions in specific communities, such as a potential CSF proteomics or CSF clinical laboratory community, and after a meta-analysis it will become clearer how to threshold bad samples (outliers) or bad studies, even for sample types which are new or uncommon to a respective lab.

Taken together, the study results clearly demonstrate the necessity of profound, multimeric data quality evaluation for the design and performance of quantitative studies. The developed universal *MaCProQC* tool allows for a preliminary evaluation study to be performed quickly and easily on both raw data and identification levels, so that the QC part does not take more than one day. We plan to adopt the *MaCProQC* tool to write the results to the standard .mzqc format and to adapt it to analyze raw data from different vendors once it is finalized (https://psidev.info/mzqc, accessed on 2 February 2022). Finally, checking the quantification quality is also highly important for all other sample types, and the *MaCProQC* tool, together with the developed R-scripts for assessing the quality of quantification and normalization, is optimally suited for this.

## Figures and Tables

**Figure 1 biomolecules-13-00491-f001:**
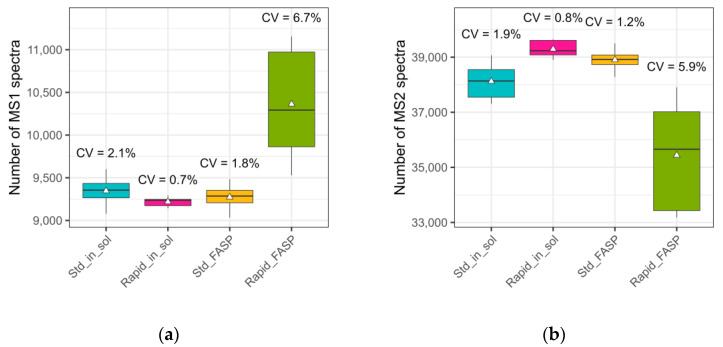
Raw data quality assessment: number of MS1 and MS2 spectra. CSF samples were digested using different approaches: Std_in_sol—standard in-solution digestion; Rapid_in_sol—rapid in-solution digestion; Std_FASP—standard filter-aided preparation; Rapid_FASP—rapid filter-aided preparation. Some 500 ng of the respective digestions were measured with nanoLC-MS and the acquired data were analyzed using our in-house developed quality control tool *MaCProQC*. (**a**) the number of precursor spectra and (**b**) number of fragment spectra as well as the coefficient of variation (CV) of these between replicates (n = 5) were analyzed. The highest variation in both the number of MS1 and MS2 spectra was observed for the rapid FASP; the lowest one for the rapid in-solution digestion.

**Figure 2 biomolecules-13-00491-f002:**
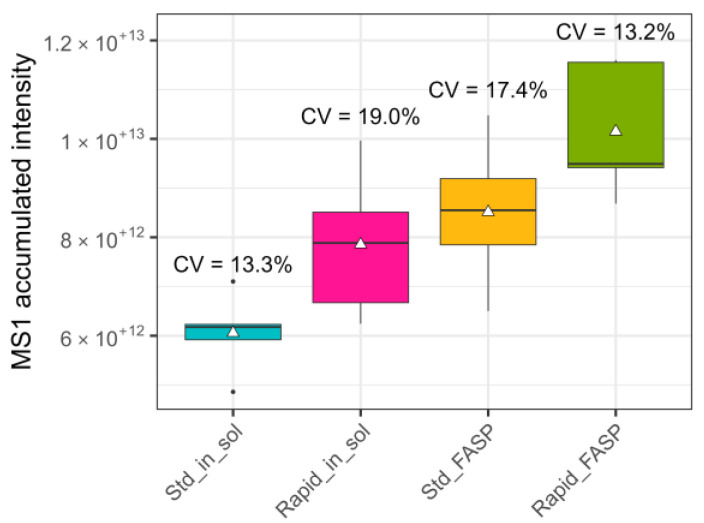
Raw data quality assessment: MS1 accumulated intensity. Accumulated TICs and the coefficient of variation (CV) for these between replicates (n = 5) were analyzed. The variation between the replicates did not exceed 20%, regardless of the sample preparation protocol. The highest CV was found for rapid in-solution digestion (19%); the lowest for standard in-solution digestion (13.3%) and rapid FASP (13.2%).

**Figure 3 biomolecules-13-00491-f003:**
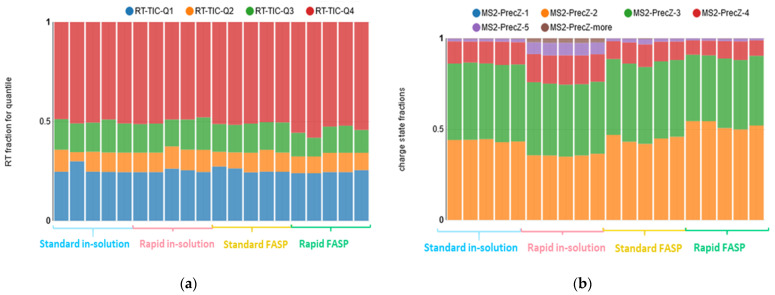
Raw data quality assessment: ion chromatogram and charge state distribution. The similarities of the obtained chromatograms were assessed in *MaCProQC* using (**a**) TIC quantile analysis, where the quantiles reflect the fraction of the retention time (RT) referring to the first (RT-TIC-Q1), second (RT-TIC-Q2), third (RT-TIC-Q3), and fourth (RT-TIC-Q4) quarter of the total run’s intensity. Slight shifts were seen for the rapid FASP. Additionally, (**b**) the distribution of the precursor charge states, i.e., +2 (MS2-PrecZ-2), +3 (MS2-PrecZ-3), +4 (MS2-PrecZ-4) etc., was examined and showed less stable distribution for both FASPs.

**Figure 4 biomolecules-13-00491-f004:**
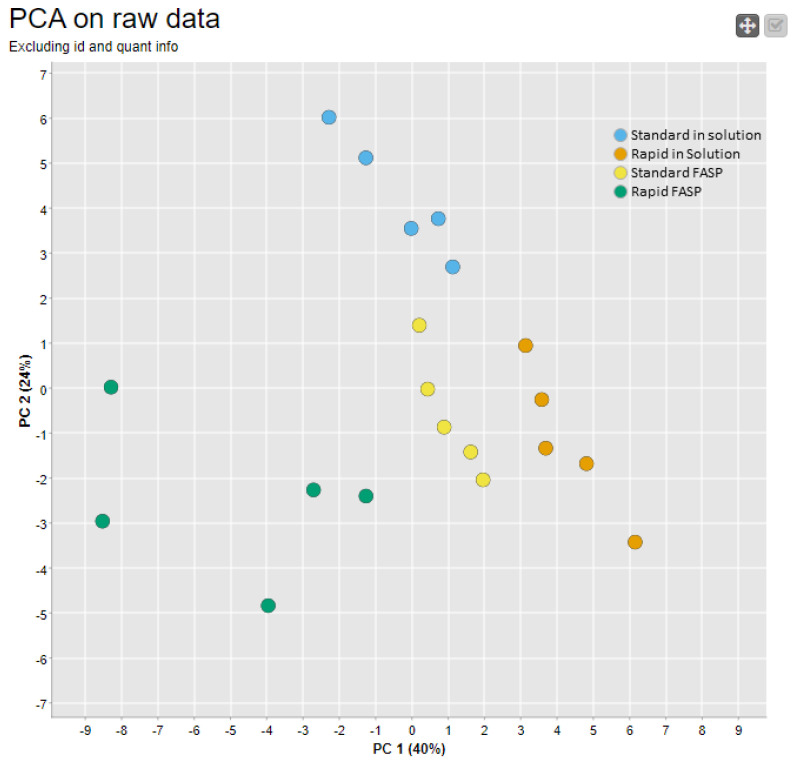
Raw data quality assessment: combined analysis of all raw data QC features using principal component analysis (PCA). CSF samples were digested using different approaches, with five replicates each for standard in-solution digestion (blue), rapid in-solution digestion (orange), standard FASP (yellow), and rapid FASP (green). Using MaCProQC, a PCA plot based on raw data features was created in order to visualize inter- and intragroup variability in the four sample datasets. PCA based on the raw data features showed the lowest level of replicate clustering for rapid FASP, testifying to the limited reproducibility of the preparation.

**Figure 5 biomolecules-13-00491-f005:**
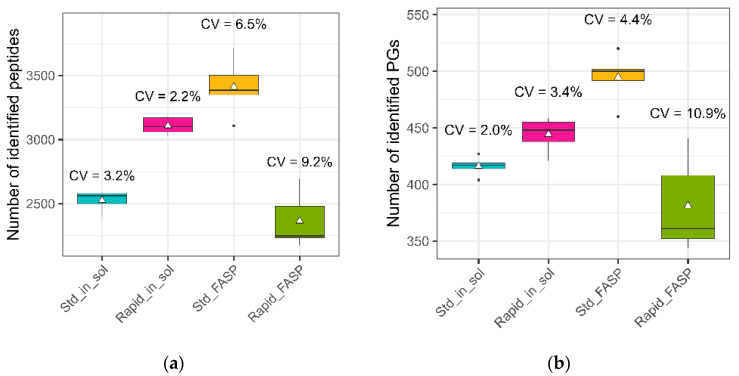
Number of identifications for the CSF samples obtained with the MaCProQC tool. The number of identified peptides (**a**) and protein groups (**b**) as well as the variation of these values between replicates were analyzed for all four sample preparation strategies. Rapid FASP showed the lowest average identification number for both peptides and protein groups, while standard FASP showed the highest, but with a higher variation compared to the in-solution digestion protocols.

**Figure 6 biomolecules-13-00491-f006:**
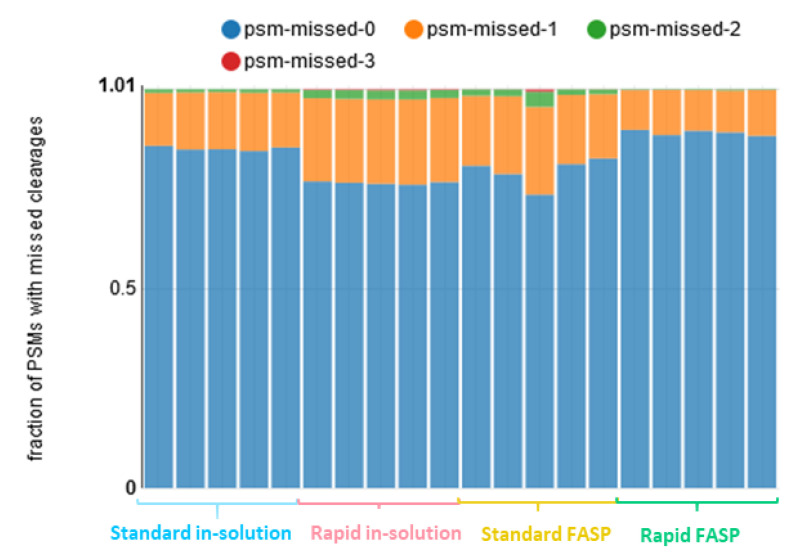
Distribution of PSMs based on missed cleavages. Distributions of the PSM fractions containing 0 (psm-missed-0), 1 (psm-missed-1), 2 (psm-missed-2), and 3 (psm-missed-3) missed cleavages were analyzed. The highest percentage of PSMs for peptides carrying missed cleavage sites was shown for the rapid in-solution digestion. Moreover, the standard FASP showed the least reproducible digestion.

**Figure 7 biomolecules-13-00491-f007:**
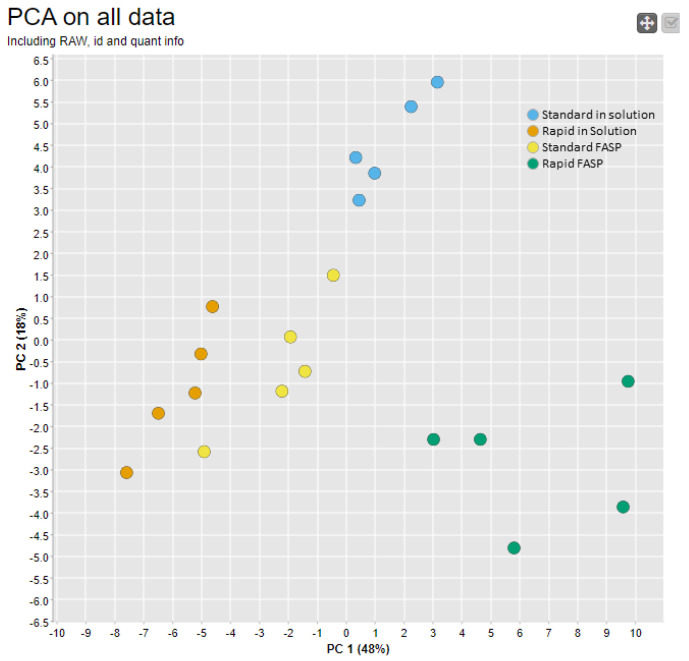
PCA analysis of the raw and identification data features. CSF samples were digested using different approaches with five replicates each for standard in-solution digestion (blue), rapid in-solution digestion (orange), standard FASP (yellow), and rapid FASP (green). Using MaCProQC, PCA based on all data (raw, identifications, and quantifications) was carried out in order to visualize inter- and intragroup variability for all four datasets. FASP clustered the worst, pointing to the lowest reproducibility.

**Figure 8 biomolecules-13-00491-f008:**
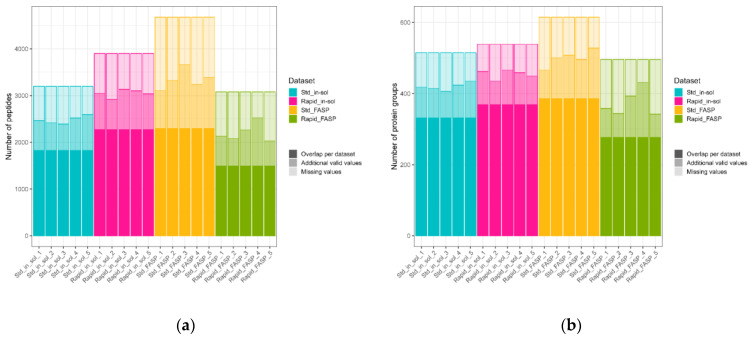
Number of quantified peptides and PGs and data completeness. Peptides (**a**) and PGs (**b**) were quantified by intensity using MaxQuant. In a dark tint, the number of consistent identifications for the corresponding dataset (overlapping for five replicates) is displayed. The dark tint and the one level lighter tint together indicate the total number of identifications in a sample. The lightest tint corresponds to the missing values in a sample. The highest data consistency was detected for the standard FASP and rapid in-solution digestion datasets. The lowest data consistency and the least reproducible results containing the most missing values were obtained for rapid FASP.

**Figure 9 biomolecules-13-00491-f009:**
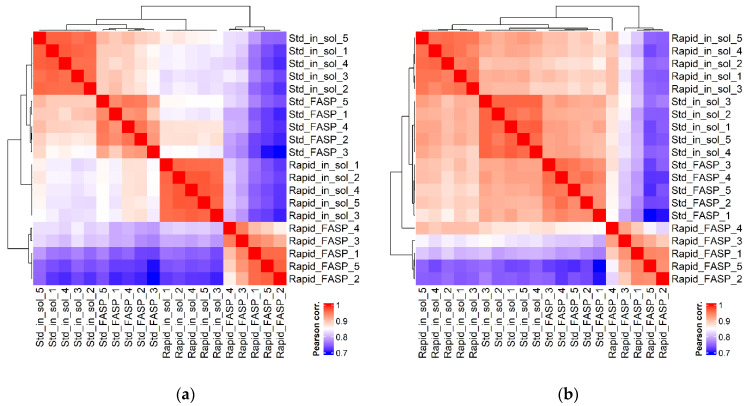
Replicate clustering based on correlation. Hierarchical clustering heatmap of Pearson correlation coefficients between dataset replicates for peptide (**a**) and PG (**b**) intensities. Correlations are colored on a scale from red (1.0, maximal possible correlation) to blue (0.7, lower correlation). Numerical values are shown in Appendix A (Correlations_peptides.xlsx and Correlations_proteins.xlsx). Std_in_sol—standard in-solution digestion; Rapid_in_sol—rapid in-solution digestion; Std_FASP—standard filter-aided preparation; Rapid_FASP—rapid filter-aided preparation. Rapid FASP was demonstrated to be the most divergent from other datasets and showed the lowest Pearson correlation coefficient between the individual replicates.

**Figure 10 biomolecules-13-00491-f010:**
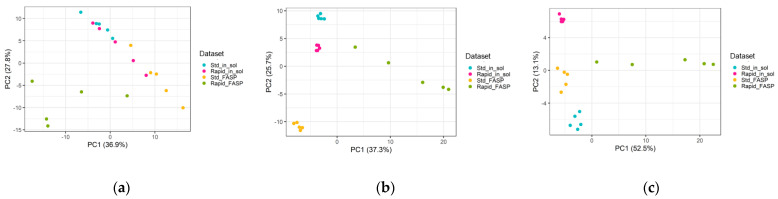
Evaluation of data normalization effectiveness. Principal component analysis (PCA) of the protein levels in different sample: (**a**) based on raw intensities; (**b**) based on LOESS normalized intensities; (**c**) based on MaxQuant’s normalized LFQ intensities. The PCA based on protein intensities showed the rapid FASP as the least reproducible. Additionally, both normalization strategies were not effective in improvement of replicate clustering (minimize the data redundancy).

## Data Availability

The mass spectrometry proteomics data have been deposited with the ProteomeXchange Consortium via the PRIDE partner repository with dataset identifier PXD037650 and 10.6019/PXD037650.

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
