# Peer review of "Quality Control—A Stepchild in Quantitative Proteomics: A Case Study for the Human CSF Proteome"

_biomolecules, 2023, doi:10.3390/biom13030491_

Round 1

Reviewer 1 Report

Letter to the authors

Reading the proposed manuscript by Svitlana Rozanova et al "Quality control - a stepchild in quantitative proteomics: a case study for the human CSF proteome" I have a real discomfort. This one is: well written, meets all the criteria in terms of publication, has no plagiarism, etc.. In this case, we have difficulty in discerning the precise message that the authors want to convey.

Is it the proteomic study of a CSF case

 or the quality controls to be set in place during a large-scale analysis

 or finally the evaluation of a MaCProQC software suite?

It is an established fact that in proteomics the reproducibility of data analysis is closely linked to data quality and that data analysis can be used to improve the robustness and reproducibility of analyses. This challenge of acquiring consistent, high-quality data in such large-scale studies is crucial and is too often hampered by variations in instrumental performance over time, as well as artifacts introduced by the samples themselves, or related to collection, storage and processing. These pitfalls have in the past often been studied and documented. Without establishing the field literature published to date, many articles shed light on the field and could have been used to support their results.

(New Guidelines for Publication of Manuscripts Describing Development and Application of Targeted Mass Spectrometry Measurements of Peptides and Proteins Susan Abbatiello et al

Quality standards in proteomics research facilities: Common standards and quality procedures are essential for proteomics facilities and their users Cristina Chiva et al

In Search of a Universal Method: A Comparative Survey of Bottom-Up Proteomics Sample Preparation Methods Gina Varnavides et al.

 In general, it is regretful that the article remains superficial in its analysis of the disparity of results observed between the different pipeline of digestion.      Figure 1: Raw data quality assessment: number of MS1 and MS2 spectra. Could we have an explanation for the variability observed for the MS1 and MS2 data for the Rapid-FASP digestion compared to the other pipelines. Since this is the same CSF sample used across workflows, what is the problem here, is it analytical, related to the sample preparation or material (membrane) used? What does the analysis of peptide sequences tell us about their physicochemical nature?

  Figure 9: the analysis of "Hierarchical clustering heatmap of Pearson between dataset replicates for peptide" shows real differences. What are the reasons for these differences? What is the nature of the peptides involved can we deduce something from this?

In figure 8, "Number of quantified peptides and PGs and data completeness" a more detailed analysis of the nature of the peptides and even more of the identified proteins would also be welcome.

  In the case of the Fast FASP approach, we observe more MS1 with a much higher intensity but less MS2 and therefore fewer identifications. Do you have an explanation for these observations (figure 1B and 5A are somewhat redundant I will switch one of the two in the additional data part). A table regrouping all the results would increase the understanding of the article for the readers.

  About comments deduced from Figure 4, can you be more explicit about the possible reasons for such a difference observed for the loosest clustering observed for the rapid FASP, "only difference being that the standard FASP showed slightly higher intergroup separation".Do you have a more detailed explanation of the reasons for these differences (poor control of the experimental conditions, nature of the peptides)? From an identification point of view, what is the recovery of the identified proteins between all these experiments?

It would be useful for the author to integrate all the data collected in the form of a table to allow a better understanding of their results. In addition, some tables could be moved in the supplementary data section. Comments and explanations regarding the differences observed at the peptide and protein levels should be provided in the sample preparation section.

More reasonably and clearly, in this article, it seems that the authors are positioning themselves in the validation of a software suite that would allow the establishment of the good progress and the quality control of MS analyses of clinical proteomics.  From this point of view, an evaluation of their workflow should be made in comparison with the existing one by mentioning what their approach brings in addition compared to the already numerous publications in the field. For example compared to :

                Proteomics Quality Control: Quality Control Software for MaxQuant Results. Bielow C et al

                Quality Control Analysis in Real-time (QC-ART): A Tool for Real time Quality Control Assessment of Mass Spectrometry-based Proteomics Data. Stanfill BA et al

                Proline: an efficient and user-friendly software suite for large-scale proteomics David Bouyssié et al.

                pmartR: Quality Control and Statistics for Mass Spectrometry-Based Biological Data, Kelly G. Stratton et al.

                QC Metrics from CPTAC Raw LC-MS/MS Data Interpreted through Multivariate Statistics Xia Wang et al.

            QCloud, (https://github.com/proteomicsunitcrg/qcloud2-pipeline)

           Skyline/Panorama/Statistical Process Control in Proteomics (SProCop)

Or compared to MaxQuant, Peaks, proteomediscoverer, , MassChroQ, OpenMS Peptide shaker 

To conclude, before publication extent the comparison to existing solutions and the contribution of MaCProQC should be established.

Reviewer 2 Report

Svitlana Rozanova et al. have developed and report here a quality control (QC) tool named MaCProQC to assess the quality of mass spectrometry-based proteomic data. They tested it on the analysis of one CSF sample and compared different sample preparation procedures. Indeed, as stated by the authors, “quality assessment of MS-data is insufficiently considered of rarely addressed in publications” or at least, it is not presented as an important part of the work before statistical analysis of the data. The quantification quality assessment should also be performed more systemically as it is the core of the data that is used in clinical research. The work is therefore of relevance and important for standardization. Still, a few revisions should be considered.

The data presented are established on replicate samples which might be considered as QC samples. The quantification assessment is based on similarity between replicates based on clustering. Quality of quantitative data between different samples is much more complex and the scatter plots the authors proposes rather not adapted in that case. The authors should discuss on how they envisage to implement their QC routine in a clinical research study. How practically this will be implemented? How will be defined the thresholds where for instance reproducibility is not sufficient for label-free proteomics? The readers truly miss recommendations from the authors. Should the use of internal standards be considered for improved quantification assessments from sample to sample?

Generally, the manuscript oscillates between the definition of QC parameters and the evaluation of different sample preparation protocols for CSF. What is the main message the authors would like to convey? Are there any conclusions on the way CSF should be handled? The readers may again miss recommendations from the authors on one side on the definition of QC thresholds and on the other side on the best protocol to use for the proteomic analysis of CSF. It appears not sufficient to conclude with: “we suggest the method of CSF digestion should be chosen based on the experimental design and the research question”.

Some graphics would benefit from further explanation. For instance, what could explain results displayed in Figure 2?

In the supplementary files, one can notice some tailing of the chromatographic peaks for the Standard FASP and Rapid FASP samples (Figure S2). Chromatography should also be considered as a QC criterion. This brings to another question that is the order of the sample injections for LC-MS/MS. Were the sample injections randomized in this study?

The number of identified peptides is taken as a criterion and is indeed important. Nonetheless, can it be considered positive criteria for method comparison as it may be linked to the presence of missed cleavages (see page 14, lines 511-512)?

Minor points:

Page 3: were the samples purified before LC-MS/MS or directly injected?

Page 4: can the authors clarify why the peptide fragments were analysed in the Orbitrap analyser and with a resolution of 30000 for this label-free approach?

Page 7, line 246: the recovery with FASP method is limited. It sounds like an important result as several automated solutions use such technology. Is there previous report available?

Page 8, line 308: the authors conclude that MS2 level plays the most important role in variations. Isn’t expected given the stochasticity of the DDA method? What are the exact parameters under the “MS2 level” (numbers of MS2 scans, precursors’ charge state or others)? What would be the picture if MS2 parameters are not considered?

Figure 6: could the authors make a link between missed cleavages (Figure 6) and the charge state distribution (Figure 3)?

 Note that I did not test the MaCProQC tool during the review process.

Round 2

Reviewer 1 Report

Hello,

The article is publishable in its current form. The authors have made the corrections and changes we requested. However, as I mentioned in a previous review, I have one comment to make. In the strictest (medical) sense, this is not a case study.  The title overstates the study and should be modified.

Yours sincerely.

Reviewer 2 Report

Overall, the authors have addressed the comments in a satisfactory manner.

Minor points:

I am still not convinced the samples should not have been randomized (they should be in a clinical study), or the MS acquisition parameters are optimal. I do trust the authors there on their previous experiences and evaluations. Maybe some references may help.

Page 12, line 437: Carbamidomethylation was considered before as a fixed modification. So, it is not clear what the authors mean in this paragraph.

Page 16, line 573 and 578: What is “carbamilization”?

Round 3

Reviewer 2 Report

No further comments. The authors have addressed the different points.